# Comparative Study of Phenotypes and Genetics Related to the Growth Performance of Crossbred Thai Indigenous (KKU1 vs. KKU2) Chickens under Hot and Humid Conditions

**DOI:** 10.3390/vetsci9060263

**Published:** 2022-05-31

**Authors:** Kitsadee Chomchuen, Veeraya Tuntiyasawasdikul, Vibuntita Chankitisakul, Wuttigrai Boonkum

**Affiliations:** 1Department of Animal Science, Faculty of Agriculture, Khon Kaen University, Khon Kaen 40002, Thailand; kitsadee_ch@kkumail.com (K.C.); veerayat@kkumail.com (V.T.); vibuch@kku.ac.th (V.C.); 2Network Center for Animal Breeding and Omics Research, Faculty of Agriculture, Khon Kaen University, Khon Kaen 40002, Thailand

**Keywords:** average daily gain, body weight, crossbreeding, genetic, indigenous chicken

## Abstract

To improve the body weight and growth performance traits of crossbred Thai indigenous chickens, phenotypic performance and genetic values were estimated. Crossbred Thai indigenous chickens, designated KKU1 and KKU2, were compared. The data included 1375 records of body weight (BW0, BW2, BW4, and BW16), breast circumference at 6 weeks of age (BrC6), and average daily gain (ADG0–2, ADG0–4, and ADG0–6). A multi-trait animal model with the average information-restricted maximum likelihood (AI-REML) was used to estimate the genetic parameters and breeding values. The results showed that the body weight and breast circumference traits (BW2, BW4, BW6, and BrC6) for the mixed sex KKU1 chickens were higher than for the KKU2 chickens (*p* < 0.05). For the growth performance traits, the KKU1 chickens had higher average daily gain and feed intake and a lower feed conversion ratio than the KKU2 chickens (*p* < 0.05). The survival rates were not different except at up to 6 weeks of age, when that of the KKU1 chickens was slightly lower. The specific combining ability, heritability, genetic and phenotypic correlations, and estimated breeding values showed that the KKU1 chickens had better genetics than the KKU2 chickens. In conclusion, KKU1 chickens are suitable for development as crossbred Thai indigenous chickens for enhanced growth performance and for commercial use.

## 1. Introduction

Indigenous chickens are very important to local economies, especially in developing and underdeveloped countries [1,2,3] and are classified as an important genetic resource in ensuring food security for countries around the world [3,4,5]. Indigenous chickens exist in all regions of the world. In Thailand, the indigenous chicken population accounted for 23% (111,855,130 birds) of the country’s total poultry population in 2021, and their production capacity was second after broilers (68%). More than 96% of the population in rural areas of Thailand own indigenous chickens [6]. At the same time, the world’s demand for poultry meat is likely to increase steadily every year [7]. From a marketing perspective, indigenous chicken meat has been proven to stand out in many respects, such as its good-quality meat, delicious taste, and use as a health food [8,9,10]. However, the main disadvantage of indigenous chickens is their slow growth [5,11,12]. As a result, it takes longer to raise them to market weight.

Growth performance traits are of primary concern in breeding programs because they are economically important traits in poultry, and these traits are under complex genetic control [1]. However, developing indigenous chickens to market competitiveness with commercial broilers may be difficult because indigenous chickens have the significant limitation of a slow growth rate. At the same time, farmers must also consider what they can do; for example, they can raise chickens in open-house systems or even free-range systems, and they can feed them with local, readily available food. Additionally, these chickens adapt well to harsh climates, especially hot and humid conditions. For these reasons, crossbreeding between indigenous and commercial chickens is an interesting approach currently [13,14]. Nevertheless, the critical question is how to breed them so that the resulting hybrid chickens have the excellent qualities of both indigenous chickens, i.e., they are easy to raise under hot and humid conditions, and broilers, i.e., they grow fast.

The Network Center for Animal Breeding and Omics Research (NCAB), Faculty of Agriculture, Khon Kaen University, Thailand, recognized the importance of crossbreeding and developed the synthetic Thai chicken line called Kaimook e-san1 (KM1: 50% Thai indigenous breed) in 2010 [1], and another synthetic Thai chicken line called Kaimook e-san2 (KM2: 25% Thai indigenous breed) in 2014 [10], with the goal of developing crossbred Thai indigenous chickens. Until 2020, they were successful in the development of crossbred chickens called KKU1 (Khon Kaen University crossbred chicken line 1) and KKU2 (Khon Kaen University crossbred chicken line 2). In addition, our previous studies on KM1 and KM2 chicken meat reported their high antioxidant substances, such as anserine and carnosine [8] with low fat compared with commercial chicken meat [15], which are beneficial to consumer health. However, assessment of the chickens’ growth performance and genetics in terms of multiple traits is also necessary before scaling up crossbreeding in the future. Additionally, the knowledge from this research may be applied in the development of crossbred indigenous chicken breeds in other regions as well. Thus, the objective of this research was to determine suitable crossbred chickens (KKU1 and KKU2) in terms of their growth performance, genetic parameters, and estimated breeding values and to provide information for genetic selection and propagation in the future.

## 2. Materials and Methods

### 2.1. Data and Animal Management

This study was conducted at the experimental farm of the Network Center for Animal Breeding and Omics Research (NCAB), Faculty of Agriculture, Khon Kaen University, in Northeast Thailand under hot and humid conditions (average air temperature of 35 °C and an average relative humidity 70% all year round). All animals used in this study were approved by the Institute of Animal for Scientific Purpose Development (No. IACUC-KKU-37/64). The data records consisted of 653 records for KKU1 (KKU1 chickens were from mating between commercial broilers (Cobb breed, n = 20) and Thai indigenous synthetic chickens (50% Thai indigenous breed called Kaimook e-san1, n = 100)) and 666 records for KKU2 (KKU2 chickens were from mating between Kaimook e-san2 (25% Thai indigenous breed, n = 20) and Kaimook e-san1 (50% Thai indigenous breed, n = 100) chickens). The parent chickens were managed intensively in a battery cage system with dimensions 60 cm × 45 cm × 45 cm and one rooster per cage in an open environment system. Artificial insemination was performed twice a week. Semen collected from one cock was used for inseminating five chicken hens. A leg band with an individual number was attached to hatched chicks to enable identification. They were raised using warming with a 100-watt lamp for 2 weeks. The lightening program consisted of two stages: the first stage was from hatching to 3 weeks with 24 h light/0 h dark; the second stage was from 3 to 6 weeks with 23 h light/1 h dark. All chickens were raised under the same environmental conditions, with open-air housing and a vaccination program. Feed was provided ad libitum in the form of a commercial broiler diet: first, a starter feed containing 21% crude protein (CP), 3100 kcal of Metabolizable Energy (ME) per kilogram, and 5% crude fiber was given to chicks aged 1 to 3 weeks; subsequently, for the growing period, the feed contained 19% CP and 3200 kcal of ME per kilogram, and this was fed from 4 weeks of age until chickens reached slaughtering weight (6 weeks of age).

### 2.2. Genetic Model and Statistical Analysis

Body weight and growth performance data were analyzed using least square means, and statistical differences were compared for synthetic chicken lines using a generalized linear model (Proc GLM) via SAS software v.9.0. The variance components, genetic parameters (heritability and genetic and phenotypic correlations), and estimated breeding values (EBVs) were estimated using the average information-restricted maximum likelihood (AI-REML) [16] with the best linear unbiased prediction (BLUP). The multi-trait animal model used in this study was as follows:Y=Xβ+Zα+e
where Y is the vector corresponding to the phenotypic values for the body weight and growth performance traits, namely birth weight (BW), body weight at 2, 4, and 6 weeks of age (BW2, BW4, and BW6, respectively), breast circumference at 6 weeks of age (BrC6), average daily gain at 0 to 2, 4, or 6 weeks of age (ADG0–2, ADG0–4, and ADG0–6, respectively); X and Z are the incidence matrices related to fixed and random effects, respectively; β is the vector of fixed effects, including the chicken hatch set and sex; α is the vector of random additive genetic effects, assumed to be $a\;\sim N(0,A\sigma_{a}^{2}$), where A is an additive relationship matrix and $\sigma_{a}^{2}$ is the additive genetic variance; e is the vector of random residual effects, assumed to be $e\;\sim N(0,I\sigma_{e}^{2}$), where I is the identity matrix and $\sigma_{e}^{2}$ is the residual variance.

The EBV estimation accuracy (Acc) can be calculated from the correlation between the estimated breeding values (EBV) and the true breeding value, from the equation $Acc=\sqrt{1-(PEV/\sigma_{a}^{2}}$), where PEV is the predicted error variance of the EBV and $\sigma_{a}^{2}$ is the additive genetic variance. 

### 2.3. Specific Combining Ability

The specific combining ability (SCA) was calculated using the following formula [17]: $SCA=\frac{1}{2}\left(X_{ij}+X_{ji}\right)-\frac{1}{2p}\left(X_{i}+X_{.j}+X_{.i}+X_{j}\right)-\frac{1}{2p}X$, where SCA is the specific combining ability effect for the crossbreeding between *i* female and *j* male parents; $X_{ij}$ is the mean for crossbred chickens resulting from crossing *i* female and *j* male parents; $X_{ji}$ is the mean for crossbred chickens resulting from crossing *j* female and *i* male parents; $X_{i}$ is the sum of the *i* female over all of the males; $X_{.i}$ is the sum of the *i* male over all of the females; $X_{j}$ is the sum of the *j* female over all of the males; $X_{.j}$ is the sum of the *j* male over all of the females; *p* is the number of breeds; and $X_{..}$ is the grand total.

## 3. Results

### 3.1. Growth Performance

The recorded data and descriptive statistics of the growth performance traits in crossbred Thai indigenous chickens (KKU1 and KKU2) are shown in Table 1. It was found that mean, minimum, and maximum values in KKU1 chickens were higher than in KKU2 chickens in all traits and the coefficient of variation (%CV) of KKU1 chickens was equal to 5.72 to 17.51%, and KKU2 chickens were equal to 7.72 to 17.33%. The results showed that KKU1 chickens were able to grow better than KKU2 chickens in open house system and climatic conditions of Thailand.

The comparisons of least square means ± standard error of the body weight and growth performance between the KKU1 and KKU2 chickens are shown in Figure 1 and Figure 2. For the body weight traits, the mixed-sex KKU1 chickens had a higher body weight at 2, 4, and 6 weeks of age (BW2, BW4, and BW6), as well as breast circumference at 6 weeks of age (BrC6), than KKU2 chickens (Figure 1a,d) (*p* < 0.05). Only birth weights were not significantly different between the KKU1 and KKU2 chickens (*p* > 0.05). The results were in the same direction in the sex-segregated analysis shown in Figure 1b,c.

As shown in Figure 2a, the ADG of the KKU1 chickens was significantly higher than that of the KKU2 chickens at all ages (*p* < 0.05). The percentage differences between the KKU1 and KKU2 chickens were as follows: 28.89%, 29.31%, and 25.66% at ADG0–2, ADG0–4, and ADG0–6, respectively. Additionally, as per the results of feed intake in Figure 2b, it was found that the KKU1 chickens had a higher feed intake than the KKU2 chickens, with a clear and significant difference between FI0–2 (24.65 g/day for KKU1 and 19.69 g/day for KKU2) and FI0–4 (45.03 g/day for KKU1 and 34.30 g/day for KKU2) weeks of age (*p* < 0.05), while the KKU1 and KKU2 chickens of FI0–6 weeks of age were not significantly different (*p* > 0.05). Regarding the feed conversion ratio (FCR) shown in Figure 2c, the KKU1 chickens had statistically significantly lower FCR values than the KKU2 chickens at all ages with the following values: 1.66, 1.89, and 2.00 at FCR0–2, FCR0–4, and FCR0–6 weeks of age for the KKU1 chickens and 1.95, 2.06, and 2.56 at FCR0–2, FCR0–4, and FCR0–6 weeks of age for the KKU2 chickens, respectively (*p* < 0.05). Figure 2d shows that the survival rate (SUR) for the KKU1 and KKU2 chickens was not significantly different (*p* > 0.05) at 0 to 2 weeks of age (SUR0–2) or from 0 to 4 weeks of age (SUR0–4) (*p* > 0.05), but this was not the case from 0 to 6 weeks of age (SUR0–6), meaning the KKU1 chickens had a significantly lower survival rate than the KKU2 chickens (*p* < 0.05). In other words, the KKU2 chickens had a lower mortality rate than the KKU1 chickens, especially in terms of the near market weight. 

### 3.2. Specific Combining Ability and Heritability Estimates

The specific combining ability (SCA), variance components, and heritability of the body weight and growth performance traits in the KKU1 and KKU2 chickens are shown in Table 2. The SCA of the KKU1 chickens was higher and positive effects for body weight and average daily gain were observed compared to the KKU2 chickens. The values of BW0, BW2, BW4, BW6, BrC6, ADG0–2, ADG0–4, and ADG0–6 was 0.21 g, 10.61 g, 32.10 g, 38.65 g, 0.37 cm, 0.75 g/day, 1.14 g/day, and 1.43 g/day for the KKU1 chickens and 0.07 g, 8.08 g, 25.26 g, 32.85 g, 0.30 cm, 0.58 g/day, 0.90 g/day, and 1.05 g/day for the KKU2 chickens. 

The heritability estimates for body weight at all ages were moderate to high in both the KKU1 and KKU2 chickens, with the highest values seen at birth and values decreasing thereafter. In addition, the heritability estimates in the KKU1 chickens were higher than those in the KKU2 chickens at all ages. The heritability estimates for body weight of the KKU1 chickens at 0, 2, 4, and 6 weeks of age (BW0, BW2, BW4, and BW6, respectively) were 0.669, 0.452, 0.375, and 0.351, while those of the KKU2 chickens were 0.634, 0.393, 0.334, and 0.336, respectively. In terms of the heritability estimates of breast circumference at 6 weeks of age (BrC6), in both KKU1 (0.304) and KKU2 (0.278) chickens, they were moderate. The heritability estimates of the average daily gain in both the KKU1 and KKU2 chickens at all ages were moderate; in the KKU1 chickens, the heritability estimates of the average daily gain at 0 to 2, 4, or 6 weeks (ADG0–2, ADG0–4, and ADG0–6, respectively) were 0.391, 0.324, and 0.276, while those in the KKU2 chickens were 0.362, 0.308, and 0.232, respectively.

### 3.3. Phenotypic Correlation and Genetic Correlation

The phenotypic and genetic correlations of body weight and growth performance in the KKU1 and KKU2 chickens are presented in Table 3 and Table 4. Generally, the results of the phenotypic correlations among the body weight traits and the average daily gain traits in both the KKU1 and KKU2 chickens were positive and ranged from low to high values. At the same time, the phenotypic correlations between the body weight and average daily gain were also positive. However, the phenotypic correlations among and between the body weight and average daily gain traits were lower than the genetic correlations. 

The genetic correlations among the body weight traits (BW0, BW2, BW4, and BW6) in both the KKU1 and KKU2 chickens were moderate to high and positive, varying from 0.25 to 0.95 for the KKU1 chickens and from 0.22 to 0.95 for the KKU2 chickens. BW2 appeared to be genetically strongly correlated with BW4 and BW6 (0.93 and 0.89 for KKU1 and 0.88 and 0.85 for KKU2). The genetic correlations among the average daily gain traits (ADG0–2, ADG0–4, and ADG0–6) were high and strongly positive in both the KKU1 and KKU2 chickens, varying from 0.80 to 0.93 for the KKU1 chickens and from 0.71 to 0.87 for the KKU2 chickens.

The genetic correlations between body weight (BW0, BW2, BW4, and BW6) and breast circumference (BrC6) were positive, and the highest values occurred between the BW6 and BrC6 traits in both the KKU1 (0.95) and KKU2 (0.92) chickens. Meanwhile, the genetic correlations between the average daily gain (ADG0–2, ADG0–4, and ADG0–6) and breast circumference (BrC6) were also strongly positive; these values were greater than 0.75 across the entire range of ADG in both crossbred Thai indigenous chickens. The genetic correlations between body weight and average daily gain in both the KKU1 and KKU2 chickens were moderate to high and positive, varying from 0.25 to 0.97 for the KKU1 chickens and from 0.21 to 0.95 for the KKU2 chickens.

### 3.4. Estimated Breeding Value

The average estimated breeding value (EBV) ± standard error and accuracy of the EBV (ACC.) of the top 20% of KKU1 and KKU2 chickens are presented in Table 5. For the body weight traits (BW0, BW2, BW4, and BW6), the average EBVs in the mixed-sex KKU1 chickens were higher than in the mixed-sex KKU2 chickens at all ages. The percentage differences between the KKU1 and KKU2 chickens were 2.16%, 3.61%, 28.52%, and 17.00% for BW0, BW2, BW4, and BW6, respectively. While the average daily gain traits of all periods (ADG0–2, ADG0–4, and ADG0–6) of both the mixed-sex chicken groups showed the same direction as the body weight traits, the KKU1 chickens had higher EBVs of ADG at all ages than the KKU2 chickens, expressed as percentage difference as follows: 6.32%, 16.67%, and 19.23% for ADG0–2, ADG0–4, and ADG0–6, respectively. For the mean EBVs of breast circumference at 6 weeks of age (BrC6), the KKU1 chickens had a higher average EBV than the KKU2 chickens, with 23.68% of the difference. Moreover, the results for the male and female chicken groups were consistent with those of the mixed-sex chickens. 

## 4. Discussion

The least square means ± standard error of body weight and growth performance compared between KKU1 and KKU2 chickens are shown in Figure 1 and Figure 2. For the body weight traits, the mixed-sex KKU1 chickens had higher body weight at 2, 4, and 6 weeks of age (BW2, BW4, and BW6), as well as breast circumference at 6 weeks of age (BrC6), than the KKU2 chickens (Figure 1a,d) (*p* < 0.05). This study aimed to investigate the body weight, growth performance, and genetic ability of two crossbred Thai indigenous chickens (KKU1 and KKU2). The body weight and weight gain of the KKU1 chickens were higher than those of the KKU2 chickens, which shows that the KKU1 chickens grew from hatching to slaughtering weight (1.2 kg) faster than the KKU2 chickens. The KKU1 chickens having a higher body weight than the KKU2 chickens also affected their breast circumference; the KKU1 chickens had a larger breast circumference than the KKU2 chickens, indicating that the body weight and breast circumference were positively correlated. The reason for this is the different genetics of the two crossbred indigenous chickens; the KKU1 chickens had 25.0% indigenous chicken blood, while the KKU2 chickens had 37.5% indigenous chicken blood. Higher levels of indigenous chicken blood were associated with a slower growth rate when compared with lower indigenous levels, which indicates that KKU1 chickens have better growth potential and are more suitable for commercial development than KKU2 chickens. When comparing the body weights of four commercial broiler breeds raised in Thailand (Cobb 500, Ross 308, Arbor Acres, and Hubbard) at 39 days of age [18], body weights were higher than those of KKU1 chickens at 42 days (6 weeks), although such differences in body weight may not be directly compared due to different farming systems, i.e., commercial broilers are raised in closed house systems while KKU1 was raised in an open house system. However, one crucial aspect of this study is that farmers can raise chickens in Thailand’s climate without relying on modern equipment or housing, which is a high cost and constraint for rural farmers. For this reason, KKU1 chickens can answer this question better than commercial broilers. However, the body weight of the KKU1 and KKU2 chickens was shown to be significantly higher compared to purebred and other crossbred indigenous chickens, while when compared to other Thai indigenous breeds, such as Chee chickens (Chee KKU12 and Chee N) [1,19], it was found to be two times higher at 6 weeks of age. In addition, when considering only KKU1 chickens, it was found that they had a higher body weight than Lueng Hang Kao Kabinburi Thai indigenous chickens [12]. Additionally, KKU1 chickens only take 6 weeks to grow from hatching to slaughtering weight, while Lueng Hang Kao Kabinburi chickens need more than 8 weeks before they can be caught and sold. When comparing the body weights of KKU1 and KKU2 chickens to those of indigenous chickens in other countries, it was found that KKU1 and KKU2 chickens have a four- and three-times higher body weight than Horro chickens in Ethiopia [20] at 6 weeks, as well as compared to local Venda chickens in South Africa [21] and Mazandaran indigenous chickens [22]. In crossbred chickens, the body weight at all ages (BW2, BW4, and BW6) of both the KKU1 and KKU2 chickens was higher than that reported in crossbred Chinese indigenous [23], crossbred Italian indigenous [24], crossbred Indian indigenous [25], and Thai synthetic chickens [1,10,11]. The differences depend on the genetics of the animals, selection program, and feed and feeding, in addition to environmental differences. The FCR showed that KKU1 chickens are more appropriate and cost-effective for commercial use in the production of crossbred indigenous chickens rather than KKU2 chickens. Moreover, the results were lower than those previously reported for other chickens, such as crossbred Nigerian indigenous [26], crossbred Korean indigenous [27], and crossbred Indian indigenous [28] chickens. In addition, the survival rate of both the KKU1 and KKU2 chickens was higher than 90%. This suggests that open-house farming is possible under hot–humid climatic conditions such as those in Thailand, where such results encourage the raising of chickens in community farming settings with budget and ventilation equipment limitations.

The specific combining ability (SCA) showed more appropriate body weights and average daily gains when crossbreeding commercial broiler and Thai indigenous chickens (Kaimook e-san1) to produce KKU1 chickens compared to crossbreeding Kaimook e-san1 and Kaimook e-san2 to produce KKU2 chickens. SCA involves dominance, overdominance, and epistasis; it also refers to the degree to which the average performance of a specific cross departs from the additive gene effect [17], and it has been used to denote the degree of nonadditive gene effect in a population. Therefore, according to the results in Table 2, KKU1 chickens are a good crossbreed for improved growth rate and growth performance traits.

For heritability, the heritability estimates of the body weight and average daily gain traits in the present study were medium to high (ranging from 0.232 to 0.669; see Table 2), similar to the results of studies carried out on local Venda chickens [21], Mazandaran indigenous chickens [22], and Thai indigenous chickens [12,29,30]. This demonstrates that genetics influence these traits to a degree that is sufficient for genetic evaluation with acceptable accuracy. The highest heritability for body weight was exhibited at two weeks of age (not including day 0 chicks), which then decreased with increasing age. Similar results were reported by Dana et al. [20], Saatchi et al. [31], and Manjula et al. [32]. The high heritability for body weight at day 0 is due to the inclusion of the maternal genetic effect [33,34]. Heritability is a value that helps in decision making in farm management planning. If the trait studied has a moderate-to-high heritability value, it means that improvement of the trait should focus on improving the genetic condition to be more cost-effective rather than improving the environment. In addition, heritability is associated with other genetic values such as selection progress and breeding value; if the traits have a high heritability value, genetic improvement in such a manner will result in very quick and accurate selection.

The estimates of genetic and phenotypic correlations among and between the observed traits varied from low to high (Table 3 and Table 4). Positive genetic correlations indicate that selection of one trait can improve the performance of other traits. The genetic correlations for body weight among the age groups were positive with moderate-to-high levels, similar to the results of studies conducted on Horro chickens in Ethiopia [20], chickens of the male Vanaraja male line [35], and the Thai crossbred black-bone chicken [36]. When considering these correlations, we suggest that growth to 2 weeks of age could be a good trait for use in a selection program, as it was highly correlated with BW4, BW6, and BrC6 (0.93, 0.89, and 0.81, respectively). The findings enable early planning to select chickens with good genetics for growth traits (selecting from two weeks after hatching) and save raising and feed costs. In addition, the genetic correlations between BW2 and the ADG0–2, ADG0–4, and ADG0–6 traits were positive (0.79, 0.69, and 0.67, respectively). Therefore, simultaneous selection for high body weight and high average daily gain traits could potentially improve both traits. Moreover, the genetic correlation between BW2 and BrC6 was also highly positive (0.81). Generally, breast circumference can be used to indicate the amount of breast meat; achieving a higher body weight will result in more breast meat. One reason for this is the pleiotropic genes associated with body weight and weight gain. For example, the growth hormone gene, in addition to being associated with an increase in body weight, can also increase weight gain [37,38,39].

The EBV showed that the KKU1 chickens, in addition to providing high phenotype growth efficiency, also had higher genetic growth efficiency than the KKU2 chickens. Using EBVs to select animals will allow the selection of animals directly from the genetic value and will be more efficient than phenotypic selection [40]. Therefore, KKU1 chickens are appropriate to develop as crossbred Thai indigenous chickens.

## 5. Conclusions

Although KKU2 had a high survival rate, the KKU1 chickens had higher phenotypic and genetic performance in all traits than the KKU2 chickens. Therefore, the utilization of indigenous chickens can be achieved in the form of crossbred chickens at a 25% indigenous blood level raised in an open-air house system to develop crossbred Thai indigenous chickens for commercial use and to promote agricultural occupations. 

## Figures and Tables

**Figure 1 vetsci-09-00263-f001:**
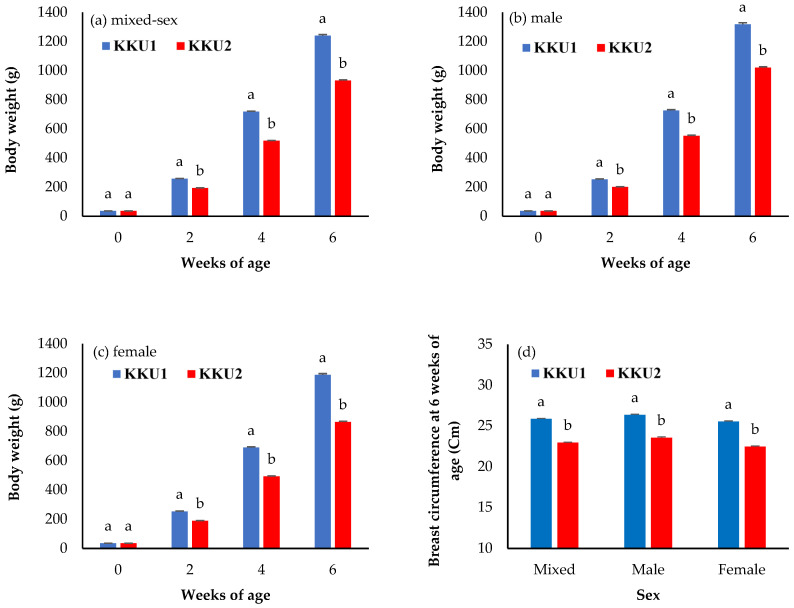
Least square means ± standard error of (**a**) the body weight of the mixed-sex chickens (birth weight (0), 2, 4, and 6 weeks of age), (**b**) body weight of the males (birth weight (0), 2, 4, and 6 weeks of age), (**c**) body weight of the females (birth weight (0), 2, 4, and 6 weeks of age), and (**d**) breast circumference at 6 weeks of age of the mixed-sex, male, and female chickens. KKU1 = crossbred chicken between commercial broiler and Thai indigenous synthetic chickens (Kaimook e-san1); KKU2 = crossbred chicken between Kaimook e-san2 and Kaimook e-san1 (both of which are Thai indigenous synthetic chickens). Means for the same trait with different letters (^a,b^) differ significantly (*p* < 0.05).

**Figure 2 vetsci-09-00263-f002:**
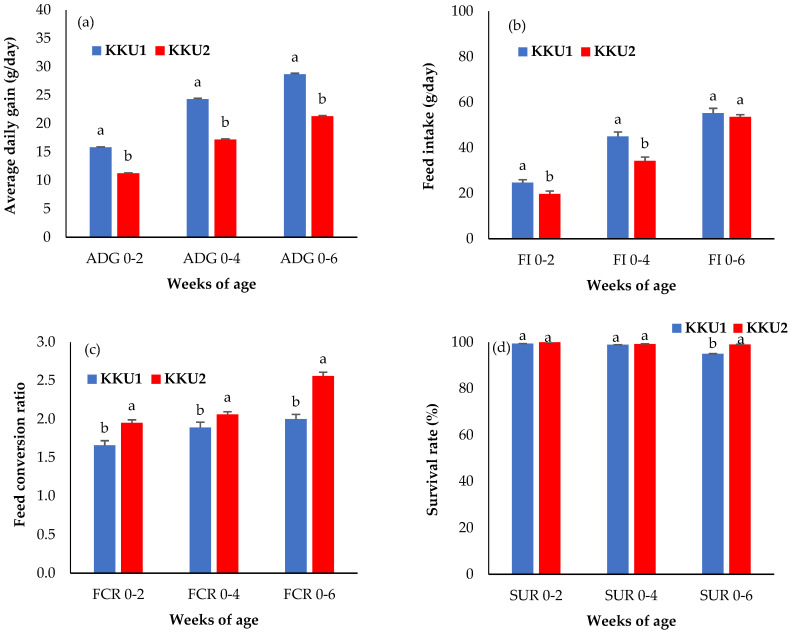
Least square means ± standard error of (**a**) the average daily gain (ADG0–2, ADG0–4, and ADG0–6), (**b**) feed intake (FI0–2, FI0–4, and FI0–6), (**c**) feed conversion ratio (FCR0–2, FCR0–4, and FCR0–6), and (**d**) survival rate (SUR0–2, SUR0–4, and SUR0–6). KKU1 = crossbred chicken between commercial broiler and Thai indigenous synthetic chickens (Kaimook e-san1); KKU2 = crossbred chicken between Kaimook e-san2 and Kaimook e-san1 (both of which are Thai indigenous synthetic chickens). Means for the same trait with different letters (^a,b^) differ significantly (*p* < 0.05).

**Table 1 vetsci-09-00263-t001:** Descriptive statistics of the growth performance traits in crossbred Thai indigenous chickens.

Traits	Number of Records	Mean	SD	Min.	Max.	CV (%)
**KKU1 chickens**						
BW0 (g)	653	36.30	4.75	26.00	52.00	13.09
BW2 (g)	646	257.75	41.30	107.00	397.00	16.02
BW4 (g)	645	717.53	100.34	329.00	978.00	13.98
BW6 (g)	541	1240.52	170.59	760.00	1800.00	13.75
BrC6 (cm)	541	25.87	1.48	19.80	30.00	5.72
ADG0–2 (g/day)	646	15.82	2.77	5.36	25.79	17.51
ADG0–4 (g/day)	645	24.33	3.54	10.43	33.57	14.55
ADG0–6 (g/day)	541	28.68	4.09	17.00	42.12	14.26
**KKU2 chickens**						
BW0 (g)	666	35.81	3.90	23.00	45.00	10.89
BW2 (g)	654	193.42	28.70	101.00	263.00	14.84
BW4 (g)	658	517.38	77.93	288.00	730.00	15.06
BW6 (g)	658	931.31	134.60	450.00	1300.00	14.45
BrC6 (cm)	656	22.94	1.77	18.00	28.20	7.72
ADG0–2 (g/day)	654	11.25	1.95	4.57	16.43	17.33
ADG0–4 (g/day)	658	17.30	2.73	9.14	24.61	15.78
ADG0–6 (g/day)	658	21.32	3.18	9.79	30.17	14.92

BW0 = birth weight (g); BW2, BW4, and BW6 = body weight at 2, 4, and 6 weeks of age (g), respectively; BrC6 = breast circumference at 6 weeks of age (cm); ADG0–2, ADG0–4, and ADG0–6 = average daily gain at 0–2, 0–4, and 0–6 weeks of age (g/day), respectively; KKU1 = crossbred chicken between commercial broiler and synthetic Thai indigenous synthetic chickens (Kaimook e-san1); KKU2 = crossbred chicken between Kaimook e-san2 and Kaimook e-san1 (both of which are Thai indigenous chickens); Mean = average value; SD = standard deviation; Min. = minimum value; Max. = maximum value; CV = coefficient of variation (%).

**Table 2 vetsci-09-00263-t002:** Variance components and heritability (standard error) of the crossbred Thai indigenous chickens.

Chickens	KKU1					KKU2				
Traits	SCA	$\sigma_{a}^{2}$	$\sigma_{e}^{2}$	$\sigma_{p}^{2}$	$h^{2}$	SCA	$\sigma_{a}^{2}$	$\sigma_{e}^{2}$	$\sigma_{p}^{2}$	$h^{2}$
**BW0**	0.21	9.58	4.73	14.31	0.669 (0.05)	0.07	8.93	5.15	14.08	0.634 (0.04)
**BW2**	10.61	457.73	554.62	1012.35	0.452 (0.07)	8.08	249.05	384.98	634.03	0.393 (0.05)
**BW4**	32.10	2698.20	4506.30	7204.50	0.375 (0.05)	25.26	1924.00	3835.00	5759.00	0.334 (0.03)
**BW6**	38.65	4094.80	7581.30	11,676.10	0.351 (0.02)	32.85	3500.50	6925.00	10,425.50	0.336 (0.04)
**BrC6**	0.37	1.33	3.04	4.37	0.304 (0.03)	0.30	1.07	2.78	3.85	0.278 (0.03)
**ADG0–2**	0.75	2.12	3.30	5.42	0.391 (0.02)	0.58	1.13	1.98	3.11	0.362 (0.02)
**ADG0–4**	1.14	3.67	7.64	11.31	0.324 (0.02)	0.90	3.17	7.12	10.29	0.308 (0.02)
**ADG0–6**	1.43	5.85	15.34	21.19	0.276 (0.01)	1.05	4.77	15.77	20.54	0.232 (0.01)

SCA = specific combining ability; $\sigma_{a}^{2}$, $\sigma_{e}^{2}$, and $\sigma_{p}^{2}$ = additive genetic variances, residual variances, and phenotypic variances, respectively; $h^{2}$ = heritability; BW0 = birth weight (g); BW2, BW4, and BW6 = body weight at 2, 4, and 6 weeks of age (g), respectively; BrC6 = breast circumference at 6 weeks of age (cm); ADG0–2, ADG0–4, and ADG0–6 = average daily gain at 0–2, 0–4, and 0–6 weeks of age (g/day), respectively; KKU1 = crossbred chicken between commercial broiler and Thai indigenous synthetic chickens (Kaimook e-san1); KKU2 = crossbred chicken between Kaimook e-san2 and Kaimook e-san1 (both of which are Thai indigenous synthetic chickens).

**Table 3 vetsci-09-00263-t003:** Phenotypic correlations (standard error) in the KKU1 (above diagonal) and KKU2 (below diagonal) chickens.

Traits	BW0	BW2	BW4	BW6	BrC6	ADG0–2	ADG0–4	ADG0–6
**BW0**	-	0.28 (0.05)	0.17 (0.05)	0.14 (0.03)	0.09 (0.04)	0.17 (0.02)	0.12 (0.02)	0.25 (0.03)
**BW2**	0.26 (0.04)	-	0.68 (0.01)	0.58 (0.00)	0.51 (0.04)	0.59 (0.01)	0.61 (0.03)	0.51 (0.03)
**BW4**	0.17 (0.01)	0.75 (0.01)	-	0.85 (0.00)	0.66 (0.01)	0.67 (0.02)	0.71 (0.01)	0.68 (0.08)
**BW6**	0.13 (0.05)	0.79 (0.02)	0.80 (0.01)	-	0.76 (0.02)	0.58 (0.01)	0.71 (0.02)	0.75 (0.09)
**BrC6**	0.10 (0.01)	0.50 (0.06)	0.64 (0.06)	0.71 (0.03)	-	0.62 (0.02)	0.73 (0.00)	0.81 (0.01)
**ADG0–2**	0.15 (0.01)	0.58 (0.01)	0.75 (0.05)	0.59 (0.05)	0.50 (0.02)	-	0.62 (0.01)	0.72 (0.01)
**ADG0–4**	0.10 (0.02)	0.65 (0.05)	0.77 (0.01)	0.79 (0.04)	0.63 (0.02)	0.56 (0.02)	-	0.75 (0.02)
**ADG0–6**	0.06 (0.01)	0.46 (0.05)	0.67 (0.05)	0.84 (0.01)	0.66 (0.01)	0.67 (0.02)	0.72 (0.01)	-

BW0 = birth weight (g); BW2, BW4, and BW6 = body weight at 2, 4, and 6 weeks of age (g), respectively; BrC6 = breast circumference at 6 weeks of age (cm); ADG0–2, ADG0–4, and ADG0–6 = average daily gain at 0–2, 0–4, and 0–6 weeks of age, respectively; KKU1 = crossbred chicken between commercial broiler and Thai indigenous synthetic chickens (Kaimook e-san1); KKU2 = crossbred chicken between Kaimook e-san2 and Kaimook e-san1 (both of which are Thai indigenous synthetic chickens).

**Table 4 vetsci-09-00263-t004:** Genetic correlations (standard error) in the KKU1 (above diagonal) and KKU2 (below diagonal) chickens.

Traits	BW0	BW2	BW4	BW6	BrC6	ADG0–2	ADG0–4	ADG0–6
**BW0**	-	0.39 (0.05)	0.29 (0.05)	0.25 (0.03)	0.32 (0.04)	0.25 (0.02)	0.24 (0.02)	0.28 (0.03)
**BW2**	0.34 (0.04)	-	0.93 (0.01)	0.89 (0.00)	0.81 (0.04)	0.79 (0.00)	0.69 (0.03)	0.67 (0.03)
**BW4**	0.25 (0.04)	0.88 (0.01)	-	0.95 (0.00)	0.96 (0.01)	0.90 (0.02)	0.94 (0.01)	0.92 (0.08)
**BW6**	0.22 (0.04)	0.85 (0.02)	0.95 (0.01)	-	0.95 (0.02)	0.91 (0.01)	0.94 (0.02)	0.97 (0.09)
**BrC6**	0.23 (0.00)	0.78 (0.06)	0.80 (0.06)	0.92 (0.03)	-	0.82 (0.02)	0.90 (0.00)	0.96 (0.01)
**ADG0–2**	0.25 (0.01)	0.72 (0.01)	0.88 (0.05)	0.85 (0.05)	0.76 (0.02)	-	0.80 (0.01)	0.89 (0.01)
**ADG0–4**	0.24 (0.01)	0.69 (0.05)	0.95 (0.01)	0.91 (0.04)	0.83 (0.02)	0.71 (0.02)	-	0.93 (0.02)
**ADG0–6**	0.21 (0.01)	0.64 (0.05)	0.92 (0.05)	0.95 (0.01)	0.97 (0.01)	0.76 (0.02)	0.87 (0.01)	-

BW0 = birth weight (g); BW2, BW4, and BW6 = body weight at 2, 4, and 6 weeks of age (g), respectively; BrC6 = breast circumference at 6 weeks of age (cm); ADG0–2, ADG0–4, and ADG0–6 = average daily gain at 0–2, 0–4, and 0–6 weeks of age, respectively; KKU1 = crossbred chicken between commercial broiler and Thai indigenous synthetic chickens (Kaimook e-san1); KKU2 = crossbred chicken between Kaimook e-san2 and Kaimook e-san1 (both of which are Thai indigenous synthetic chickens).

**Table 5 vetsci-09-00263-t005:** Estimated breeding value ± standard error (S.E.), and accuracy of the estimated breeding value of the top 20% Thai indigenous synthetic chickens.

Chickens	Mixed-sex	ACC.	Male	ACC.	Female	ACC.
KKU1						
BW0	4.25 ± 0.01	90.61	4.27 ± 0.01	92.68	3.80 ± 0.01	90.38
BW2	13.76 ± 0.03	73.30	14.89 ± 0.03	82.19	12.62 ± 0.05	73.36
BW4	35.55 ± 0.03	65.23	36.94 ± 0.05	74.05	31.39 ± 0.06	65.08
BW6	55.62 ± 0.07	63.05	65.39 ± 0.04	65.36	44.71 ± 0.05	71.33
BrC6	0.47 ± 0.03	75.31	0.48 ± 0.05	77.59	0.40 ± 0.05	73.27
ADG0–2	1.85 ± 0.06	72.27	1.90 ± 0.07	75.60	1.75 ± 0.04	72.68
ADG0–4	2.10 ± 0.06	72.21	2.14 ± 0.05	73.07	1.99 ± 0.05	72.00
ADG0–6	1.55 ± 0.04	68.11	1.59 ± 0.06	70.83	1.48 ± 0.08	66.84
**KKU2**						
BW0	4.16 ± 0.01	86.76	4.10 ± 0.01	87.19	3.65 ± 0.01	86.25
BW2	13.28 ± 0.07	69.59	14.29 ± 0.05	69.70	12.75 ± 0.06	69.76
BW4	27.66 ± 0.08	62.89	29.52 ± 0.05	62.66	25.09 ± 0.08	63.00
BW6	47.54 ± 0.04	67.68	52.82 ± 0.02	67.36	39.93 ± 0.04	67.89
BrC6	0.38 ± 0.03	65.45	0.39 ± 0.03	65.28	0.36 ± 0.04	65.67
ADG0–2	1.74 ± 0.04	68.31	1.72 ± 0.04	68.46	1.56 ± 0.06	68.56
ADG0–4	1.80 ± 0.05	62.92	1.91 ± 0.04	62.92	1.77 ± 0.07	62.57
ADG0–6	1.30 ± 0.06	64.18	1.33 ± 0.05	64.22	1.29 ± 0.08	64.30

BW0 = birth weight (g); BW2, BW4, and BW6 = body weight at 2, 4, and 6 weeks of age (g), respectively; BrC6 = breast circumference at 6 weeks of age (cm); ADG0–2, ADG0–4, and ADG0–6 = average daily gain at 0–2, 0–4, and 0–6 weeks of age, respectively; KKU1 = crossbred chicken between commercial broiler and Thai indigenous synthetic chickens (Kaimook e-san1); KKU2 = crossbred chicken between Kaimook e-san2 and Kaimook e-san1 (both of which are Thai indigenous synthetic chickens); ACC. = accuracy of the estimated breeding value.

## Data Availability

The data presented in this study are available on request from the Network Center for Animal Breeding and Omics Research, Faculty of Agriculture, Khon Kaen University, Thailand.

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
