# Peer review of "Comparative Study of Phenotypes and Genetics Related to the Growth Performance of Crossbred Thai Indigenous (KKU1 vs. KKU2) Chickens under Hot and Humid Conditions"

_vetsci, 2022, doi:10.3390/vetsci9060263_

Round 1

Reviewer 1 Report

Following are my comments:

Introduction:

  • The sentence on Line 31 which states “Indigenous chickens are presently distributed in all regions of the world.” does not make sense to me. Replace it with “Indigenous chickens exist in all regions of the world.” Or any other more appropriate sentence. The term “distributed” is more appropriate in the context of a single breed of chicken that is distributed around the world.  
  • The sentence on the lines 61-66 is too long. Make it short and correct its grammar. Words like “important substances” should be replaced with some other better word.
  • The grammar of sentences on lines 85-90 can be improved.
  • Replace “Lighting was divided into two stages” with “the lightening program consisted of two stages” or any other more appropriate sentence.

Materials and Methods:

  •  Table 1. should be presented in the results section.
  • On line 110 replace “variance component” with “variance components”.
  • On line 116 it has been mentioned that “Y is the vector corresponding to the phenotypic values for the body weight and …”. If the applied model is a multi-trait model then how can Y be a vector? 
  • On lines 122-123 there are errors in the sentence “vector of random additive genetic effects, assumed to be ? ~?(0, ??2?), and ?2 a is the additive genetic variance;”. The term ? ~?(0, ??2?) should be corrected to represent additive genetic variance.
  • On line 126 the sentence “The estimation of the EBV accuracy (Acc) can be calculated” can be replaced by “The EBV estimation accuracy (Acc) can be calculated”.

  • To evaluate the commercial use of these crosses as mentioned on line 25 of the Abstract, it would have been much better to see the comparison of the two crosses with commercial broiler or any other commercial breed.  However, now I think it is better to present such a comparison in the Discussion section based on the literature.

Results:

  • Table 1 should be provided in the Results section and should be explained along with Figures 1 and 2 for a better representation of the results.
  • I also think that variation within these traits as presented in Table 1 is also worth mentioning in the Results section because for commercial use of these crosses, this property is also very important.
  • In Table 2 additive and error variance values have been provided which do not add any valuable information. I think providing phenotypic variance along with heritability for all the traits should be enough.
  • I think it is better to first present the phenotypic correlations and then the genetic correlations.

Discussion:

  • On line 308 remove indigenous from the sentence to read “Higher levels of indigenous chicken blood were associated with”
  • A multi-trait model was used without giving any reasoning for it. Furthermore, I am unable to see the information that EBVs add to this manuscript.

Author Response

Response to Reviewer 1 Comments

Introduction:

Point 1: The sentence on Line 31 which states “Indigenous chickens are presently distributed in all regions of the world.” does not make sense to me. Replace it with “Indigenous chickens exist in all regions of the world.” Or any other more appropriate sentence. The term “distributed” is more appropriate in the context of a single breed of chicken that is distributed around the world.

Response 1: We agree with you, we therefore replaced the sentence from “Indigenous chickens are presently distributed in all regions of the world” to “Indigenous chickens exist in all regions of the world” as your suggestion. See lines 31-32 in the revised MS.

Point 2: The sentence on the lines 61-66 is too long. Make it short and correct its grammar. Words like “important substances” should be replaced with some other better word.

Response 2: We have revised the mentioned sentence more precisely according to your suggestion. See line 61-64 in the revised MS.

Point 3: The grammar of sentences on lines 85-90 can be improved.

Response 3: We have revised the mentioned sentence more precisely according to your suggestion. See lines 84-89 in the revised MS.

Point 4: Replace “Lighting was divided into two stages” with “the lightening program consisted of two stages” or any other more appropriate sentence.

Response 3: We have revised the mentioned sentence more precisely according to your suggestion. See line 89 in the revised MS.

Materials and Methods:

Point 5: Table 1. should be presented in the results section.

Response 5: Table 1 is moved to present in the results part as your suggestion.

Point 6: On line 110 replace “variance component” with “variance components”.

Response 6: it was replaced as your suggestion. See line 104 in the revised MS.

Point 7: On line 116 it has been mentioned that “Y is the vector corresponding to the phenotypic values for the body weight and …”. If the applied model is a multi-trait model then how can Y be a vector?

Response 7: I assure you that Y must be in the form of column vector of observation on animals for traits (t) i; i = 1 to t. If written in the form of Henderson's Mixed Model Equation (Henderson, 1973) will be as follows:

  •  
  • ;

Reference:

Henderson, C.R. 1973.  Sire evaluation and genetic trends.  pp 10-41.  In Proceeding of The Animal Breeding and Genetics Symposium in Honor of Dr. Jay L. Lush.  American Society of Animal Science, IL.

Point 8: On lines 122-123 there are errors in the sentence “vector of random additive genetic effects, assumed to be ? ~(0, ??2?), and ?2 a is the additive genetic variance;”. The term ? ~(0, ??2?) should be corrected to represent additive genetic variance.

Response 8: We have adjusted according to your suggestion. See lines 115-117 in the revised MS.

Point 9: On line 126 the sentence “The estimation of the EBV accuracy (Acc) can be calculated” can be replaced by “The EBV estimation accuracy (Acc) can be calculated”.

Response 9: the sentence was replaced as your suggestion. See line 120 in the revised MS.

Point 10: To evaluate the commercial use of these crosses as mentioned on line 25 of the Abstract, it would have been much better to see the comparison of the two crosses with commercial broiler or any other commercial breed.  However, now I think it is better to present such a comparison in the Discussion section based on the literature.

Response 10: We added the discussion between crossbred Thai native chicken with broiler chicken in Thailand. See Discussion part in lines 333-341.

Results:

Point 11: Table 1 should be provided in the Results section and should be explained along with Figures 1 and 2 for a better representation of the results.

Response 11: Table 1 is moved to present in the results part as your suggestion. In addition, we explained the results along with the Figure 1 and 2. See in lines 153-159.  

Point 12: I also think that variation within these traits as presented in Table 1 is also worth mentioning in the Results section because for commercial use of these crosses, this property is also very important.

Response12: We added the sentences in the result part. See in lines 153-159.

Point 13: In Table 2 additive and error variance values have been provided which do not add any valuable information. I think providing phenotypic variance along with heritability for all the traits should be enough.

Response13: We apologize for not explaining the variances of additives and errors in the results section. The presentation of these two values ​​will help breeders use them as benchmarks for selection and genetic improvement for the next generation of animal genetics. For example, it emphasizes more significant additive variability while controlling for less non-genetic variance. We still want to show these two values in Table 2.

Point 14: I think it is better to first present the phenotypic correlations and then the genetic correlations.

Response14: We rearranged the presentation by first present the phenotypic correlations and then the genetic correlations as you suggestion. See lines 238-245.

Discussion:

Point 15: On line 308 remove indigenous from the sentence to read “Higher levels of indigenous chicken blood were associated with”

Response15: Thank you for your suggestion. We have rewritten the sentences as follow.

Point 16: A multi-trait model was used without giving any reasoning for it. Furthermore, I am unable to see the information that EBVs add to this manuscript.

Response16: We added some reasons for using A multi-trait model in the introduction part. See lines 64-66. Moreover, we added the information of EBV values in both result and discussion parts. See lines 282-295 and 411-415.

Reviewer 2 Report

This research investigated the body weight, growth performance, and genetic ability of two crossbred Thai indigenous chickens. And it can provide basic information to prove KKU1 chickens are suitable for development as crossbred. However, according to the description in the discussion part KKU1 chickens had 25.0% indigenous chicken blood, while the KKU2 chickens had 37.5% indigenous chicken blood. I think the conclusion can be inferred without research.

  1. Line 33. In Thailand, …… accounted for 23% (111,855,130 birds) of the country's total. Thailand should be a province of China.
  2. Line 34. What is the meaning of “and their production capacity was second only to that of broilers”?
  3. Line 82-85. How the KKU1 and KKU2 were formed? The detail information for crossing mode should be provided. Besides, how many males and females in each group?
  4. Figure 2. You did not separate the sex. Whether it was reasonable?
  5. Line 139. What is the division standard for moderate and high heritability?
  6. Line 261-262. Why you selected the top 20% of KKU1 and KKU2 chickens to estimat the breeding value?

Author Response

Response to Reviewer 2 Comments

This research investigated the body weight, growth performance, and genetic ability of two crossbred Thai indigenous chickens. And it can provide basic information to prove KKU1 chickens are suitable for development as crossbred. However, according to the description in the discussion part KKU1 chickens had 25.0% indigenous chicken blood, while the KKU2 chickens had 37.5% indigenous chicken blood. I think the conclusion can be inferred without research.

Response 1: The preliminary hypothesis for genetics is that KKU1 chicken (25.0 indigenous chicken blood) may produce better yields than KKU2 chicken. However, we need to know the exact figure of the observed value. (parameters) changed in both chickens. We are also interested in environmental adaptation, reflected in feed consumption, feed conversion ratio, and survival rate to be used as an academic reference and compare with chickens that may be developed in the future. Considering the animal's blood levels alone may not be sufficient or research is lacking.

Point 1: Line 33. In Thailand, …… accounted for 23% (111,855,130 birds) of the country's total. Thailand should be a province of China.

Response 1: Thailand is not a province of China. Thailand is a country in the southeast of Asia.

Point 2: Line 34. What is the meaning of “and their production capacity was second only to that of broilers”?

Response 2: The production capacity of broilers is the biggest (68%), while native chicken is second after broilers (23%). We have corrected the sentence. See lines 33-34.

Point 3: Line 82-85. How the KKU1 and KKU2 were formed? The detail information for crossing mode should be provided. Besides, how many males and females in each group?

Response 3: We added information for crossing mode and number of males and females used in KKU1 and KKU2 chickens. See lines 79-84.

Point 4: Figure 2. You did not separate the sex. Whether it was reasonable?

Response 4: In fact, under practical use, there is not easy to determine the sex individually of a day-old chick. Therefore, the hatched chicks were raised together without sex separately; concurrently, separate diets could not provide in this case as well.

Thus, the determination of ADG, FCR, and feed intake, as represented in Figure 2, could not separate the sex.    

However, the growth performance of each sex was demonstrated in Figure 1 as those chicks were attached with the leg band. 

Point 5: Line 139. What is the division standard for moderate and high heritability?

Response 5: the heritability is usually categorized into three classes: low (<0.19), moderate (0.2-0.39), and high (above 0.4) as given by Chansawang (1987); Duangjinda (2005); Supakorn, (2013)

References:

Chansawang, S. Animal breeding. Department of Animal Science, Faculty of Agriculture Kasetsart University: Bangkok, Thailand. 1987.

Duangjinda, M. Animal Genetic Evaluation. Department of Animal Science, Faculty of Agriculture, Khon Kaen University: Khon Kaen, Thailand. 2005.

Supakorn, China. Animal breeding. Bachelor of Science Program in Animal Technology and Innovation, Walailak University: Nakhon Si Thammarat, Thailand. 2013.

Point 6: Line 261-262. Why you selected the top 20% of KKU1 and KKU2 chickens to estimate the breeding value?

Response 6: In livestock farms, parental generations are typically selected as replacement herds for the next generation based on the selection from the top 20% of the animals with the highest genetics. This is because a more intensive selection of parental generations, such as those with the highest genetics, is selected. 5-10% will boost the chances of a faster blood clotting rate and the risk of gene loss, which does not benefit the next generation of animals. On the other hand, if many animals are selected for use as a replacement herd, such as 30-40% of genetic progression delays, thereby affecting numbers. 20% is the most appropriate. Some references were also attached:

                References

In dairy cattle: J. Dairy Sci. 92:4604–4612 doi:10.3168/jds.2008-1513

In pig: J. Anim. Sci. 94:5004–5013, https://doi.org/10.2527/jas.2016-0820

In chicken: Animals 2022, 12(3), 335; https://doi.org/10.3390/ani12030335

Round 2

Reviewer 1 Report

Thank you very much for making all these changes.

Author Response

dear reviewer

We are very grateful for the critical reading and your efforts to improve the quality of the manuscript. 

Sincerely yours,

Wuttigrai 

Reviewer 2 Report

I am satisfied with most of the responses. There still exsits some mistakes.

1.     Line 34. Indigenous chicken did not belong to broilers?

2.     Line 86-122. The description was repeated.

3.     Line 31. “Indigenous chickens existare presently distributed in all regions of the world”. The authors did not revise the sentence as stated. The present sentence was wrong.

Author Response

dear reviewer

   We are very grateful for the critical reading and your efforts to improve the quality of the manuscript.  We hope the responses to each comment as attached file will please you.

Sincerely yours,

Wuttigrai
